

# Role of echocardiography in screening for portopulmonary hypertension in liver transplant candidates: a meta-analysis

Xin Yin[1,2], Yueming Shao[1], Yu Zhang[1], Hui Gao[3], Tingting Qin[1], Xiaoyu Wen[1] and Chen Yang[4]

[1] Department of Hepatology, The First Hospital of Jilin University, Changchun, Jilin Province, China
[2] Chengdu Women's and Children's Central Hospital, Chengdu, Sichuan Province, China
[3] Department of Digestive System, The Hospital of Tai'an Municipal, Tai'an, Shandong Province, China
[4] Department of Bone and Joint Surgery, The First Hospital of Jilin University, Changchun, Jilin Province, China

## ABSTRACT

**Objectives**. To demonstrate the screening value of echocardiography for portopulmonary hypertension (POPH) in liver transplant candidates.

**Design**. Systematic review and meta-analysis.

**Background**. POPH is a complication of end-stage liver disease that adversely affects the outcome of orthotopic liver transplant. There are no specific symptoms in the early stage of POPH. POPH reduce the survival rate of patients with end-stage liver disease specially if they are not diagnosed. Therefore, early detection may improve prognosis. The objective of this study is to explore the screening value of echocardiography on liver transplant candidates for screening of POPH compared to right heart catheterization (RHC).

**Method**. PubMed, EMBASE and the Cochrane Library were searched by two independent reviewers for potentially eligible studies published up to 30 June 2019 to retrieve data based on per-patient analysis. STATA, Meta-DiSc, and RevMan were applied to perform this meta-analysis.

**Results**. Our search yielded 1576 studies, of which 11 satisfied the inclusion criteria. The pooled sensitivity, specificity, positive likelihood ratio (PLR), negative likelihood ratio (NLR) and area under the summary receiver operating characteristic (SROC) curve (AUC) of echocardiography for POPH were 0.85 (95% CI [0.65–0.94]), 0.83 (95% CI [0.73–0.90]), 4.99 (95% CI [3.03–8.21]), 0.19 (95% CI [0.07–0.46]), and 0.91 (95% CI [0.88–0.93]), respectively. Deeks' funnel plot did not indicate the existence of publication bias ($P = 0.66$).

**Conclusions**. Echocardiography, a noninvasive modality, provides superior screening for POPH, but the diagnosis of POPH still requires RHC. **PROSPERO registration number CRD42019144589**.

Corresponding authors
Xiaoyu Wen, 15804301609@163.com
Chen Yang, 15043031980@163.com

## INTRODUCTION

Portopulmonary hypertension (POPH) is a type of pulmonary hypertension (PH) associated with portal hypertension, which is a rare complication of end-stage liver

disease. In patients with portal hypertension, the occurrence of PH is reported to be 2% to 6% (*Budhiraja & Hassoun, 2003*). The prevalence of POPH in patients who are candidates for liver transplant varies between 3% and 10% (*Chen et al., 2013*; *Hua et al., 2009*). A robust diagnosis of POPH requires the presence of portal hypertension and hemodynamic instability upon invasive right heart catheterization (RHC), namely, a mean pulmonary artery pressure (mPAP) $\geq$ 25 mmHg, pulmonary vascular resistance (PVR) >240 dynes s cm$^{-5}$ and pulmonary capillary wedge pressure (PCWP) <15 mmHg (*Krowka et al., 2016*). Female sex and autoimmune hepatitis are associated with an increased risk of POPH, whereas hepatitis C infection is a protective factor against POPH (*Kawut et al., 2008*).

The mean survival of POPH patients is approximately 15 months without medical intervention (*Le Pavec et al., 2008*). To improve survival, patients with POPH should be treated with medication or undergo liver transplantation. However, a multicenter study reported that the mortality rate of patients with POPH is as high as 36% after liver transplantation (*Krowka et al., 2004*). Because the prevalence of POPH is relatively low, it is not feasible for all patients to undergo invasive RHC. Furthermore, 60% of the patients have no obvious clinical symptoms when POPH is diagnosed (*Hadengue et al., 1991*). Accordingly, there is a need for noninvasive methods to screen patients for POPH as early as possible. Estimated pulmonary artery systolic pressure (ePASP) on echocardiography is determined using the modified Bernoulli equation: ePASP (mmHg) = 4$\times$ TRV$^2$ + estimated right atrial pressure, with TRV representing the tricuspid regurgitant peak velocity (*Martin et al., 2014*). The risk of PH is increases when indicated by echocardiography that the pulmonary artery is widened and right heart morphology is altered (*Galie et al., 2016*). However, RHC should be performed to confirm the existence of POPH when the ePASP of a liver transplantation candidate is more than 50 mmHg (*Krowka et al., 2016*). The existence of POPH can be excluded when the ePASP, as measured by echocardiography, is less than 30 mmHg (*Raevens et al., 2013*). There have been many studies conducted on echocardiography as a screening tool for POPH, but each study reported different conclusions.

Therefore, we performed this meta-analysis, which may complement existing studies, to evaluate the accuracy of echocardiography compared to RHC as a screening method in liver transplantation patients.

## MATERIAL AND METHODS

### Search strategy

This meta-analysis was registered in PROSPERO, and the registration number was CRD42019144589. PubMed, EMBASE and the Cochrane Library were searched by two independent reviewers (Xin Yin and Yueming Shao) for potentially eligible studies published up to 30 June 2019. The search terms were a combination of medical subject headings (MESH) and keywords. The search strategy was as follows: (''Portopulmonary hypertension'' or ''porto pulmonary hypertension'' or ''POPH'' or ''PPH'' or ''PPHTN'') and (''echocardiography''). The ''All fields'' category was used for search. The retrieval

strategies were adjusted according to the different databases and were confirmed after many pre-retrievals of the combination of words above. In addition, to check for potential studies, we scanned the references list of existing systematic reviews and meta-analyses relevant to our study.

## Inclusion and exclusion criteria

All retrieved articles were screened by two independent reviewers (Xin Yin and Yueming Shao) according to the inclusion and exclusion criteria, and disagreement was resolved by a third author (Yu Zhang). The studies were required to meet the following criteria: (1) patients included in the studies were liver transplantation candidates who underwent echocardiography and RHC before liver transplantation; (2) the results of RHC served as a reference standard for diagnosis and severity; (3) a certain cut-off values for echocardiography was adopted to screen POPH patients; (4) effective tricuspid regurgitation was demonstrated by echocardiography and pulmonary artery pressure was estimated; and (5) the extracted data were available to calculate true positive, false positive, false negative and true positive values. The exclusion criteria of this study were as follows: (1) non-English articles; (2) case reports, conference abstracts, reviews, editorial materials, letters, and comments; and (3) studies involving the individuals.

## Data extraction and quality assessment

Two independent reviewers (Xin Yin and Hui Gao) extracted the following information: first author, year, sample size, mean/median age, the number of POPH/non-POPH patients, cut-off value, false negative, false positive, true negative, true positive. Disagreement was solved by discussion and if necessary, a third reviewer (Tingting Qin) was involved to reach a consensus. Quality assessment was assessed by two independent researchers (Xiaoyu Wen and Chen Yang) using Quality Assessment of Diagnostic Accuracy Studies 2 (QUADAS-2).

## Statistical analysis

The threshold effect was considered first. The $P$-value of the Spearman correlation coefficient was used to measure the threshold effect. A $P$-value greater than 0.05 indicated that there was no threshold effect and that further exploration into whether heterogeneity was caused by a non-threshold effect was needed. The heterogeneity was evaluated by the value of the $I$-square statistic using the "midas" command based on a bivariate model of a hierarchical receiver operating characteristic (HSROC). The combined sensitivity, specificity, positive likelihood ratio (PLR), negative likelihood ratio (NLR), and their 95% confidence intervals (CIs) were calculated and graphically shown using forest plots. A summary receiver characteristic curve (SROC) was applied to assess the screening accuracy of echocardiography, and the AUC was computed; the higher the AUC, the higher the screening value was. Deeks' funnel plot asymmetry test was applied to assess publication bias. This meta-analysis was conducted by STATA software (version 15.0, StataCrop, College Station, Texas, USA) and Meta-DiSc 1.4 (*Zamora et al., 2006*). Quality assessment was performed using Review Manager 5.3. $P < 0.05$ was considered to be statistically significant. Additionally, we combined the Pearson's correlation coefficient

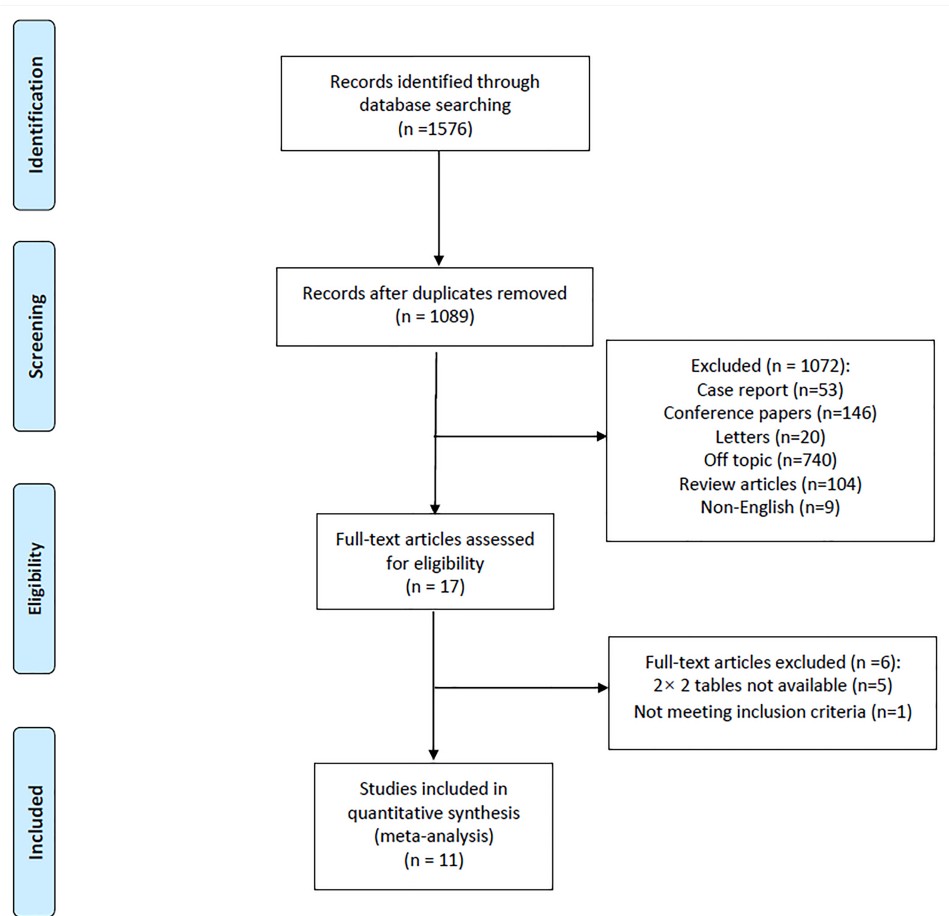

**Figure 1  Flow diagram of included studies.**

of echocardiography and RHC in liver transplantation individuals using the "metacor" package of R software (version 3.5.3).

## RESULTS

### Literature search

A total of 1,089 articles were retrieved by electronic search after duplicates were excluded. Based on the inclusion and exclusion criteria, 11 full-text articles (*Hua et al., 2009*; *Raevens et al., 2013*; *Pilatis et al., 2000*; *Habash et al., 2018*; *Colle et al., 2003*; *DesJardin et al., 2006*; *Al-Harbi et al., 2014*; *Torregrosa et al., 2001*; *Cotton et al., 2002*; *Devaraj et al., 2014*; *Farzaneh-Far et al., 2008*) were ultimately included in the meta-analysis. A PRISMA flow diagram of the retrieved studies is shown in Fig. 1.

### Basic characteristics and quality assessment

In conclusion, 11 studies published between 2000 and 2019 involving 1,160 liver transplant candidates were used in the analysis; 5 of the studies were from Europe, 4 were from the USA, and 2 were from Asia. Eight of the included studies were prospective design, and the

**Table 1  Characteristics of studies included in meta-analysis.**

| First author | Year | Sample size | Mean/ Median age | POPH/ Non-POPH | Cut-off value | TP | FP | FN | TN |
|---|---|---|---|---|---|---|---|---|---|
| Pilatis ND | 2000 | 55 | 48 | 8/47 | PASP > 40 mmHg | 5 | 1 | 3 | 46 |
| Raevens S | 2013 | 152 | 58 ±11 | 7/145 | PASP > 38 mmHg | 7 | 26 | 0 | 119 |
| Habash F | 2018 | 31 | 57 ± 11 | 17/14 | PASP > 47 mmHg | 10 | 3 | 7 | 11 |
| Colle IO | 2003 | 165 | 48 ± 8 | 10/155 | PASP > 30 mmHg | 10 | 7 | 0 | 148 |
| DesJardin JT | 2019 | 97 | 56.8 ± 8.8 | 11/86 | PASP ≥ 40 mmHg | 10 | 45 | 1 | 41 |
| Saner FH | 2006 | 74 | 49.6 ± 11.6 | 14/60 | PASP > 40 mmHg | 9 | 14 | 5 | 46 |
| AlHarbi A | 2014 | 248 | 49 ± 13.9 | 4/244 | PASP ≥ 40 mmHg | 4 | 57 | 0 | 187 |
| Hua R | 2009 | 105 | 49.5 ± 11.8 | 4/101 | PASP > 30 mmHg | 4 | 18 | 0 | 83 |
| Torregrosa M | 2001 | 107 | 57 ± 8 | 5/102 | PASP ≥ 40 mmHg | 4 | 9 | 1 | 93 |
| Cotton CL | 2002 | 78 | 51 ± 9.6 | 11/67 | PASP ≥ 50 mmHg | 6 | 10 | 5 | 57 |
| Devaraj A | 2014 | 48 | 54 | 5/43 | PASP ≥ 40 mmHg | 5 | 18 | 0 | 25 |

Notes.

POPH, portopulmonary hypertension; FN, false negative; FP, false positive; TN, true negative; TP, true positive.

Except for the last one is median age, others are mean age.

remaining 3 studies were retrospective studies. The cut-off value of echocardiography in these studies varied from 30 mmHg to 50 mmHg. Table 1 presents the detailed information of the included studies. The results of the quality assessment suggested that the risk of bias was low and that the quality of the included studies was high (Fig. S1).

## Meta-analysis

The 11 eligible studies were pooled for the present meta-analysis of diagnostic tests. In the threshold analysis, the Spearman correlation coefficient was 0.210, and the $P$-value was 0.536, indicating that there was no threshold effect. Figure 2 shows the forest plots of sensitivity, which ranged from 0.55 to 1.00 (pooled, 0.85; 95% CI [0.65–0.94]), and specificity, which ranged from 0.48 to 0.98 (pooled, 0.83; 95% CI [0.73–0.90]). The combined PLR was 4.99, and the combined NLR was 0.19, and these data are presented in the (Fig. S2). Figure S3 shows the screening odds ratio, which ranged from 5.24 to 415.80 (pooled, 26.90; 95% CI [8.37–86.40]). The AUC for echocardiography in patients who underwent liver transplant was 0.91 (95% CI [0.88–0.93]) (Fig. 3).

There were 3 articles that reported data regarding the correlation between echocardiography and RHC results in liver transplantation patients (*Farzaneh-Far et al., 2008*; *Taleb et al., 2013*; *Krowka et al., 2006*). We combined the Pearson's correlation coefficients (which ranged from 0.30–0.78) using the random-effect model and the pooled r was 0.59 (95% CI [0.20–0.79]) (Fig. 4). We could not extract the data or analyze the correlation between the two methods for POPH patients who were diagnosed by echocardiography. There were only two articles that reported Pearson's correlation coefficients between the two measurement methods. Habash (*Habash et al., 2018*) reported that there was a very poor correlation ($r = 0.58$, $P = 0.006$), as determined by the Spearman rant correlation test, between ePASP and pulmonary artery systolic pressure by RHC. In the other two articles (*Chen et al., 2013*; *Murray, Carithers Jr & AASLD, 2005*), Pearson correlation coefficients, which where $r = 0.60$ and $r = 0.75$, were used to show the

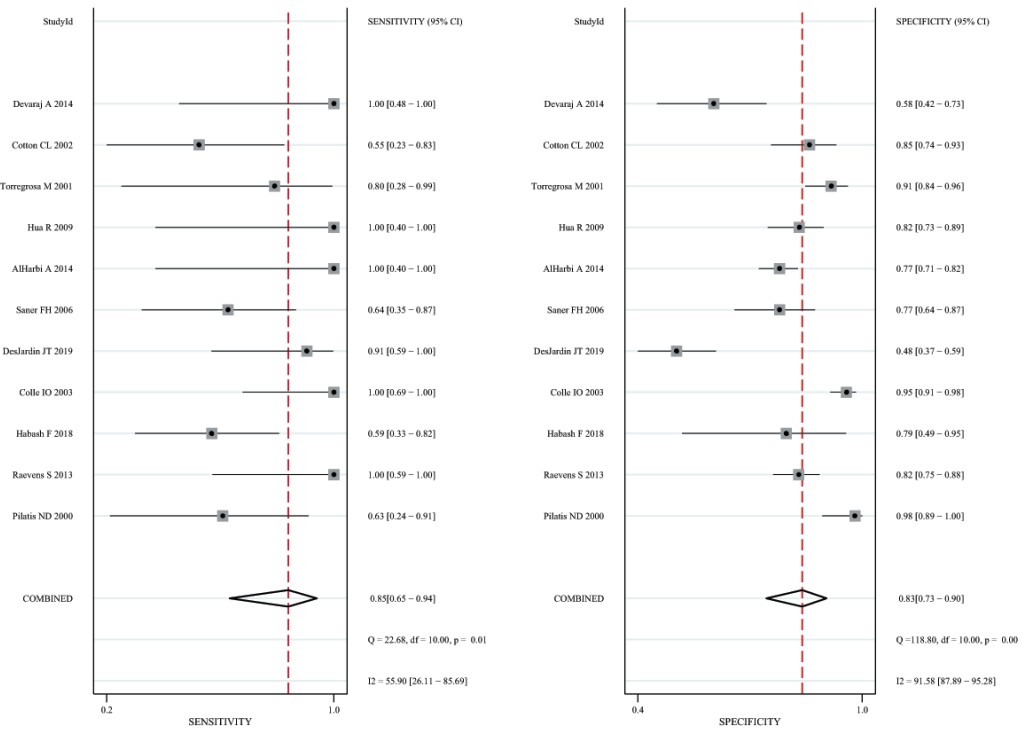

**Figure 2** Forest plot of the combined sensitivity and specificity of echocardiography for screening for POPH in liver transplant candidates.

relationship between the two measurement methods. Based on the current research results, the results of the two methods are correlated in liver transplantation patients and portopulmonary hypertension patients screened by echocardiography, but the degree of correlation still needs to be confirmed by further research.

## Publication bias

Deeks' funnel plot was applied to evaluate publication bias, which is reflected by the symmetric shape of the funnel plot, and the analysis is presented in Fig. 5. The *P*-values was 0.66, indicating that there was no significant publication bias.

## DISCUSSION

The purpose of echocardiography screening before liver transplantation is to identify patients with clinically significant POPH before surgery and improve their prognosis. Because of the invasiveness of this procedure and coagulation disorders in patients with end-stage liver disease, RHC cannot be used as a screening tool for liver disease patients. The guidelines of the American Association for the Study of Liver Disease (AASLD) suggest that all patients who are waiting for liver transplantation should be screened by echocardiography (*Janda et al., 2011*). However, there have been no reports regarding the accuracy of echocardiography as a screening tool for POPH.
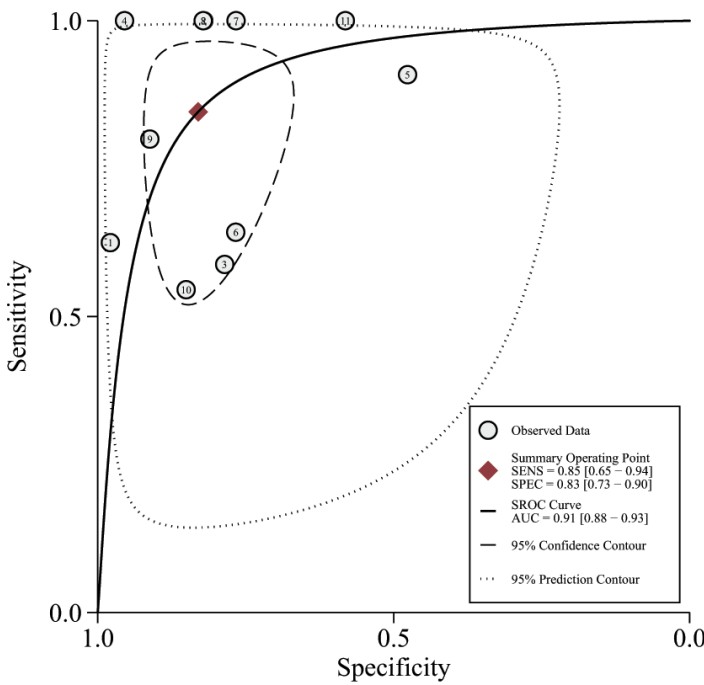

**Figure 3** The summary receiver characteristic curve of the 11 included studies.

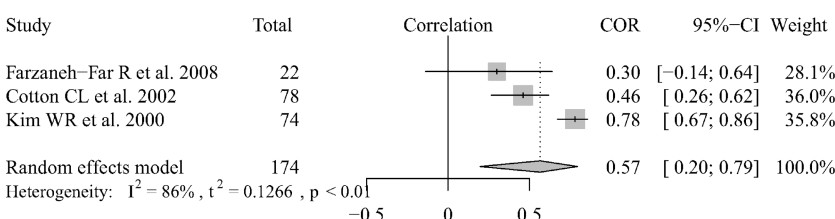

**Figure 4** Forest plot of the correlation between echocardiography and RHC.

Our study confirms that the screening accuracy of echocardiography for POPH is clinically acceptable, showing a sensitivity of 0.85 (95% CI [0.65–0.94]), specificity of 0.83 (95% CI [0.73–0.90]), and area under the SROC curve of 0.91 (95% CI [0.88–0.93]). Two meta-analyses were previously conducted to evaluate the diagnostic value of echocardiography in pulmonary hypertension (PH). A meta-analysis by de Surinder showed that the estimated sensitivity and specificity of echocardiography for patients with PH were 83% and 72% (*Posteraro et al., 2006*), respectively. Mohammed et al *Krowka et al. (2006)* conducted a meta-analysis of 9 studies among patients with PH and found that echocardiography had a sensitivity of 88% and a specificity of 56% for PH patients. However, these two studies were meta-analyses based on echocardiography as a diagnostic tool for patients with PH. Our article looked at a special group of patients with PH and is a comprehensive study of reports up to 31 June 2019. Our results showed

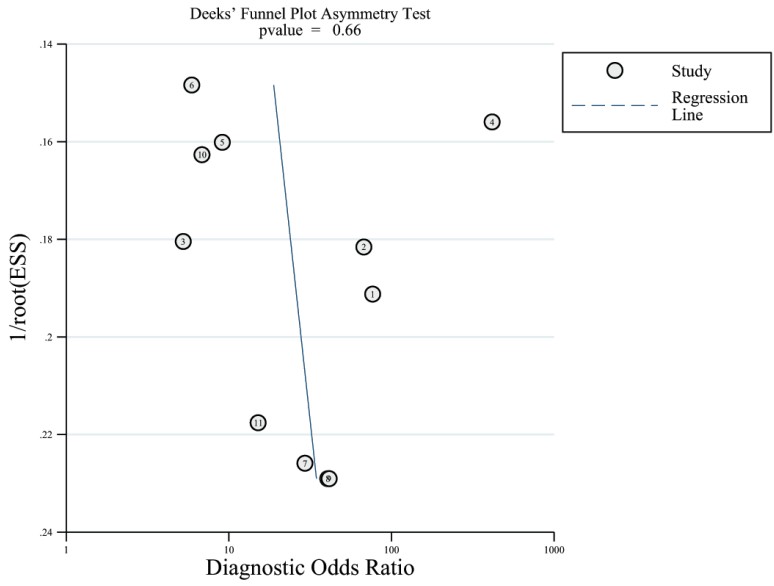

**Figure 5    Deeks' funnel plot for evaluating publication bias.**

that echocardiography had high sensitivity and specificity for detecting POPH in liver transplantation candidates.

Approximately 9.5%–22% of liver transplant candidates have no detectable tricuspid regurgitation (TR) by Doppler echocardiography (*Hua et al., 2009*; *Murray, Carithers Jr & AASLD, 2005*). However, TR is nearly invariable in patients with PH (*Garg & Armstrong, 2013*). Moreover, the specificity of echocardiography as a screening tool for POPH was 0.83 according to 11 studies. Therefore, the risk of missing POPH because of the absence of TR might be extremely low (*Colle et al., 2003*). Although our results further confirmed the screening accuracy of echocardiography in POPH patients, echocardiography does not differentiate between precapillary and postcapillary PH (*Simonneau et al., 2019*). Therefore, RHC is necessary to characterize the specific hemodynamic patterns. Importantly, these patterns may require different therapeutic approaches based on volume status. In addition, the latest guidelines (*Yock & Popp, 1984*) indicate that an mPAP of 20 mmHg should be considered the upper limit of normal. Hence, the sensitivity of echocardiography diagnosis for POPH may be underestimated by existing studies.

There were several implicit limitations in our meta-analysis. First, we included only studies published in PubMed, EMBASE and the Cochrane Library, and we excluded abstracts, letters to the editor and articles written in languages other than English. This may have led to publication bias. Second, the time interval between echocardiography and RHC was different. The longer the period between echocardiography and RHC, the higher the chance that the hemodynamic status of patients will change. Third, 1 of the 11 articles estimated right atrial pressure based on a fixed value of 10 mmHg (*Torregrosa et al., 2001*). In other studies, right atrial pressure was estimated using the inferior vena cava diameter. The use of the jugular venous pressure for clinical estimates does not allow

reliable measure of right atrial pressure and is less satisfactory than using a fixed value of 14 mm Hg to predict pulmonary artery pressure (*DesJardin et al., 2019*). Therefore, we think that the use of a fixed value of 10 mmHg has little effect on the value of pulmonary artery systolic pressure. All of the above factors increase the heterogeneity of the studies. In our study, the heterogeneity was high, but the generality of this conclusion may be affected by the absence of grouping basis for a subgroup analysis and the inability to conduct further related subgroup analysis. Consequently, our conclusions need to be interpreted with caution.

## CONCLUSIONS

In summary, echocardiography is a highly sensitive tool for noninvasive screening of POPH. However, if the echocardiography results are abnormal, RHC should be performed to confirm the diagnosis. Our study provides a basis for echocardiography as a POPH screening tool. Moreover, further larger prospective studies are recommended to verify the comprehensive effectiveness of echocardiography as a noninvasive means for detecting patients with POPH.

## ACKNOWLEDGEMENTS

Thanks to all medical workers in the hepatology department!

### Funding

The study was sponsored by the Foundation of Science and Technology Commission of Jilin Province (grants No. 20190201065JC). The funders had no role in study design, data collection and analysis, decision to publish, or preparation of the manuscript.

### Grant Disclosures

The following grant information was disclosed by the authors:
Foundation of science and Technology Commission of Jilin Province: 20190201065JC.

### Competing Interests

The authors declare there are no competing interests.

### Author Contributions

- Xin Yin and Yueming Shao performed the experiments, analyzed the data, prepared figures and/or tables, and approved the final draft.
- Yu Zhang, Hui Gao analyzed and Tingting Qinthe data, authored or reviewed drafts of the paper, and approved the final draft.
- Xiaoyu Wen and Chen Yang conceived and designed the experiments, authored or reviewed drafts of the paper, and approved the final draft.
## Data Availability

The raw measurements are available in File S1.

## Supplemental Information

Supplemental information for this article can be found online at http://dx.doi.org/10.7717/peerj.9243#supplemental-information.

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
