# Peer review of "Role of echocardiography in screening for portopulmonary hypertension in liver transplant candidates: a meta-analysis"

_PeerJ, doi:10.7717/peerj.9243_

## Round 0.1 · original submission · Major Revisions

It is an interesting study, and please provide point-by-point response to address all of the comments. Specifically, two reviewers suggested the conclusion needs to be revised and the method would be a screen tool instead of diagnosis. In the end of manuscript, you should not have two conclusion paragraphs. Furthermore, the English writing needs to be improved.

Reviewer 1 ·

Basic reporting

The article structure and language is confusing. For example, "POPH is rarely in clinic" and "POPH patients can happen to acute right ventricular failure".

Experimental design

Study is well designed but adds little to the field.

Validity of the findings

I disagree with conclusion that echocardiography can be used as a diagnostic tool as it is only a screening test. The findings from the study do not support the conclusions.

Additional comments

The introduction and discussion could be condensed for clarity and reviewed for grammar in order to be more readable.

Reviewer 2 ·

Basic reporting

Introduction can be improved
In lines 26, 27 and 28 I suggest to point that POPH reduce the survival rate of patients with end stage liver disease specially if they are not diagnosed. Early detection may improve the prognosis.

In line 28: I think it is better in this way: The objective of this study is to explore the diagnostic value of echocardiography on liver transplant candidates for diagnosis of POPH compared to RHC.

The conclusions in line 40 – 42 need to be clarified.
Since the objective of this study is not to find a cut – off value of RSVP (well defined in the literature), this is not really a limitation of the study.

Re write lines 56 and 57, can be described in more appropriate words.


Regarding lines 62- 64, echocardiogram is the gold standard for screening PAH, so you have to re think the way you want to propose the relevance of this study.

In line 68. The proper way to point out the criteria for performing a RHC is “when the PASP is more than 50mmHg”, instead of “is not less tan 50”.

In line 70 it is not necessary the comment after the reference.

Comments for the first paragraph of discussion
RHC can be safely performed in POPH patients,
There are well defined echocardiographic criteria for screening.
There is a general agreement between experts that echo is not accurate in the diagnosis of PAH, specially POPH. Again I would suggest to change the arguments for this study.

There is a lot of text without references in discussion.

Line 206,207. RCTs are not used to determine the efficacy of a diagnostic test.

Experimental design

Research question is valid.
Are you exploring the value of echocardiogram as a diagnostic tool or as a screening method?
If you review the table 1, all the studies have a lot of False Positives, this is not unexpected considering this echocardiogram is a screening method.
It is very risky to conclude that echocardiogram may substitute the RHC for diagnosis of POPH (or other types of PAH)

Validity of the findings

There is an heterogeneous cut-off value of PSAP in the studies, with your findings you cant draw a valid and robust conclusion. Also consider observations in methodology.

Additional comments

It is tempting to substitute the RHC with echocardiogram in practice, but there is bunch of data against that, all the studies you showed have a lot of false positives. Remember there are several hemodinamic profiles in cirrhotic patients (Pseudo pulmonary hypertension, fluid overload and true PAH), that can only be identified correctly by RHC.
So you have to be very careful in your conclusions.

·

Basic reporting

Use of English language needs to be checked again by the authors. Alternatively, a native English speaker should review the manuscript. Some examples where language can be improved include lines 26 (first sentence), 33, 56, 70, 95, 146, 160, 179, 183, 188, 211.

References are consistent, but need to be transformed according to the journal guidelines.

Figure legends need to be more explanatory for the reader.

Experimental design

The meta-analysis examined the diagnostic accuracy of pulmonary artery systolic pressure evaluation using TR signal and right atrial estimated pressure, not the diagnostic accuracy of echocardiography in general as other indices to evaluate pulmonary artery pressure and to differentiate precapillary and post-capillary pulmonary hypertension have also been proposed. (DesJardin et al Clinical Transplantation 2019, Dimitroglou et al Journal of Clinical Medicine 2019). Therefore in my opinion, the title of the manuscript should be more specific. Moreover, please discuss the need to evaluate alternative echocardiography indices for the diagnosis of POPH.

Validity of the findings

In the study by Colle et al, true negatives were 148 according to the authors. However, as authors clearly state in their manuscript, in 102 of 146 patients, TR jet was absent or inadequate. While, TR signals is commonly insufficient in patients with low ePASP, how the authors imply that those patients were real true negatives?

In the study by Al-Harbi et al right atrial pressure in 10mmHg for all patients. Therefore, not only ePASP values, but also method to calculate ePASP differs among studies. Please discuss this limitation.

Additional comments

It was a pleasure to read the meta-analysis by Yin et al in a topic with increasing clinical interest. Below there are some specific comments.

Abstract
Lines 28-30. Objective of the meta-analysis has already been written above.
Introduction
Line 63. While bleeding risk is cirrhotic patients is indeed higher, thrombosis is also more frequent and INR is not linearly related to coagulation function. Therefore, in my opinion the word ‘decrease’ needs to be replaced.
Meta-analysis
Lines 146-149. Should be rewritten giving number of spearman coefficient, as the reader cannot understand what the authors imply.
Discussion
Lines 192-194. Authors stated that they did not manage to identify all studies meeting the inclusion criteria. Which studies were not included? Could a different search approach identify those studies?

---

## Round 0.2 · accepted · Accept

Thank you for submitting this interesting study to us.

·

Basic reporting

I congratulate the authors for the improvement in english language.

Literature is much more informative and discussion is more explanatory.

Figure legends have been improved.

Results are relevant to the hypothesis.

Experimental design

I thank the authors for their explanation.

Validity of the findings

Results are valid and conclusions are more clear. The way limitations are presented has been improved.

Additional comments

This version is much better and in my opinion should be accepted for publication.